# An Evaluation of the Yangtze River Economic Belt Manufacturing Industry Level of Intelligentization and Influencing Factors: Evidence from China

Decai Tang [1,2], Luxia Wang [2,*] and Brandon J. Bethel [3]

[1] China Institute of Manufacturing Development, Nanjing University of Information Science & Technology, Nanjing 210044, China; tangdecai@nuist.edu.cn

[2] School of Management Science and Engineering, Nanjing University of Information Science & Technology, Nanjing 210044, China

[3] School of Marine Sciences, Nanjing University of Information Science & Technology, Nanjing 210044, China; 20195109101@nuist.edu.cn

[*] Correspondence: 20191224014@nuist.edu.cn

**Abstract:** Over recent decades, the application of artificial intelligence methods in manufacturing has led to new spheres of research such as the Internet of Things, Cyber–Physical Systems, and Cloud Computing and Big Data, leading to the so-called Industry 4.0. However, to date, little research has been geared towards assessing the factors that influence intelligent manufacturing on a regional scale. Addressing this problem, this paper constructs an evaluation index system for the Yangtze River Economic Belt (YREB) intelligent manufacturing sector using eleven years (2008–2018) of provincial panel data. The entropy method is applied to three evaluation criteria, namely intelligent innovation, equipment, and profit, to construct an evaluation index system. An analysis of the results revealed that the level intelligentization of the manufacturing industry of the YREB increases yearly, and that intelligent innovations are notably occurring at a faster rate than profits. Disproportional enterprise returns on investment have occurred, which decreases enterprise motivation to be innovative in the first place. Additionally, it was also observed that FDI, financial development, government intervention, and the level of opening-up were the primary factors modulating regional intelligent manufacturing levels.

**Keywords:** artificial intelligence; intelligentization; influencing factors; manufacturing industry; Yangtze River Economic Belt

## 1. Introduction

The Yangtze River Economic Belt (YREB) comprises nine provinces (Anhui, Guizhou, Hubei, Hunan, Jiangsu, Jiangxi, Sichuan, Yunnan, and Zhejiang) and two municipalities (Chongqing and Shanghai), sits on a total land area of over two million square kilometers, and accounts for approximately 40% of China's GDP. Consequently, in terms of China's overall economic development, the health of the YREB regional economy is crucial to maintain national stability and growth. Presently, the manufacturing industry of the YREB remains labor-intensive with high proportions of labor capital input, which in conjunction with rising labor costs and the dearth of scientific and technological innovation capacity works, jointly inhibits industrial profits. Consequently, the transformation and upgrading of the current manufacturing sector to an intelligent manufacturing sector is urgently required [1]. The earliest explanation of the concept of intelligent manufacturing can be traced back to Joseph Harington's introduction of the term "computer integrated manufacturing" in 1973, but it did not attract much attention at the time. It was not until 1984 that Harington published his book *Understanding the Manufacturing Process: Successful Ways to Implement CAD/CAM* that the American Society of Engineers paid attention to [2,3]. Subsequently, Swinbanks and Anderson proposed the drawbacks of the development

of intelligent manufacturing and the social problems brought about by it [4]. With the emergence of new technologies such as industrial Internet, big data, and cloud manufacturing, more conceptual studies related to intelligent manufacturing have emerged. Intelligent manufacturing refers to both the manufacturing process and the life cycle of a product. It also emphasizes the application of a new generation of information and communication technology and artificial intelligence. These in turn are embodied in the application of artificial intelligence to the production operation system to make it capable of self-perception, self-decision-making, and self-execution. The core of intelligent manufacturing lies in "intelligence". By contrast, intelligentization of the manufacturing industry refers to the process in which the manufacturing industry realizes intelligence through continuous efforts to promote intelligent manufacturing through the application of artificial intelligence in the redesigning of the manufacturing process and a product's lifecycle [5]. Consequently, Thoben et al. suggested that intelligent manufacturing is a corresponding system that applies information technology to achieve timely and efficient implementation at the workshop level and above [6]. Davis et al. believe that intelligent manufacturing is to optimize product production and transaction as the goal and use advanced information and manufacturing technology to improve the flexibility of the manufacturing process to cope with the dynamic global market [7]. The research on intelligent manufacturing in China began in the 1990s, and the intelligent manufacturing industry originated from digital manufacturing, experienced the stage of networked manufacturing, and gradually transitioned to the era of intelligent manufacturing [5]. Cheng believes that the weak foundation of China's manufacturing industry led to a long time span in the initial stage of China's manufacturing intelligence. To develop an intelligent primary manufacturing industry, technological innovation can be realized through scale expansion, technology introduction, and imitation learning [8]. Di and Zhang emphasized that the key to realizing the intelligent transformation and upgrading of the manufacturing industry lies in the development and application of intelligent software [9]. However, Wang and Tao believe that the connotation of manufacturing intelligence is not limited to equipment input and technology application but places more emphasis on the economic and social benefits brought by it [10].

In China, it has been identified that overall research on the intelligence level of the manufacturing industry may ignore the differences and connections between regions and industries. This has led to a bifurcation of research, where some scholars focus on specific areas for regional research, while others conduct industry research for specific industries. For example, Han et al. held that the Chinese northeast suffered from critical problems such as the absence of key core and generic technologies; the need to strengthen the collaborative innovation mechanisms between industries, universities, and research institutes; the slow green operation process of the manufacturing sector; and the lack of a collaborative transformation of the industrial chain [11]. Fu et al. evaluated the Yangtze River Delta by using the improved TOPSIS evaluation model, which combines weight with the entropy weight method, and found that innovation, quality, and expertise were the main factors affecting the high-quality development of the manufacturing industry [12]. Liu et al. found that stronger similarity in industrial structure can improve the relationship between cities and bring advantages through economies of scale [5]. There are still obstacles to regional integration and coordinated development due to the inconsistency of development levels and development goals among cities in the YREB. Additionally, Ran suggested that the coupling coordinated development level of logistics and manufacturing industries in the YREB presents a trend of "higher in the east and lower in the west", the relative effectiveness of logistics in promoting the development of manufacturing industry is low, and the relative lag of logistics development leads to the lack of sustainable development of the manufacturing industry [13]. Based on the ecological symbiosis theory, Jiang et al. constructed a YREB city-symbiotic model and another that measured the asymmetry between it and an equipment manufacturing industry symbiosis model. In a comparison of the two models, it was determined that in terms of stability and expansibility, the

mutualism model was better than the partial symbiotic model. It was also observed that the larger the symbiotic coefficient, the greater city equipment manufacturing sales output limited growth [1]. Ma et al. argued that under the backdrop of Industry 4.0, advanced technologies would be able to provide numerous opportunities for the intelligent development of an energy-intensive manufacturing industry and proposed sustainable data-driven intelligent manufacturing, which provides a new solution and an efficient decision-making way for the implementation of sustainable intelligent manufacturing [14].

Presently, while there has been an abundance of research performed into the intelligentization of the manufacturing industry in a wide array of specific regions and industries, to our knowledge, no systematic research has been conducted on the topic in the YREB. In this paper, we construct an evaluation index system for the Yangtze River Economic Belt (YREB) intelligent manufacturing sector using eleven years (2008–2018) of provincial panel data. An entropy weight method is used to analyze the level of intelligentization and its influencing factors in the YREB. Motivation for this inquiry starts with the "Made in China 2025" strategy, which was adopted to promote the transformation and upgrading of the manufacturing sector and industrial structure based on innovation, intelligent transformation, foundation strengthening, and green development [14]. The rest of the paper is structured as follows: Section 2 describes the research methodology and data. Section 3 introduces the variables considered in this research. Section 4 presents analysis results, and Section 5 concludes.

## 2. Research Methodology and Data

### 2.1. Entropy Weight Method

The entropy weight method is a commonly used objective weighting method in multi-index comprehensive evaluation, with entropy values generally being used to determine the degree of dispersion among indices. Here, the smaller the entropy value, the greater the degree of dispersion and thus, the greater the influence of a particular index on a comprehensive evaluation system [15–17]. By using this method to determine index weights, a targeted index can be objectively evaluated and analyzed, and subjective influences caused by human decision-making can be avoided. The specific process is as follows. Standardization is achieved by the estimation of positive Equation (1) and negative indicators Equation (2):

$$X_{ij} = \frac{x_{ij} - min(x_{1j}, \ldots, x_{ij})}{\max(x_{1j}, \ldots, x_{ij}) - min(x_{1j}, \ldots, x_{ij})} \tag{1}$$

$$X_{ij} = \frac{\max(x_{1j}, \ldots, x_{ij}) - x_{ij}}{\max(x_{1j}, \ldots, x_{ij}) - \min(x_{1j}, \ldots, x_{ij})} \tag{2}$$

where $X_{ij}$ and $x_{ij}$ are the standardized and original values of the *j*th index in the *i*th sample, respectively, and $i = 1, 2, 3, \ldots, n$ and $j = 1, 2, 3, \ldots, m$.

Entropy is measured as follows:

$$p_{ij} = \frac{X_{ij}}{\sum_{i=1}^{n} X_{ij}} i = 1, \ldots, n; j = 1, \ldots, m \tag{3}$$

$$e_j = -\frac{1}{\ln(n)} \times \sum_{i=1}^{n} p_{ij} \ln p_{ij}; 0 \le e \le 1 \tag{4}$$

Entropy weights are calculated as:

$$D_j = 1 - e_j \tag{5}$$

$$W_j = \frac{D_j}{\sum_{j=1}^{m} D_j}; j = 1, \ldots, m \tag{6}$$

When each indices' weights are estimated, corresponding quantitative values of the indices can be obtained.

### 2.2. Multiple Linear Regression Model

The factors affecting the intelligence level of the manufacturing industry are influenced by both small- and large-scale factors. The small-factors are mainly limited by the development level of enterprises, while the large-factors are not only restricted by the industrial development but are also related to the economic society [18,19]. This paper mainly explores the influence of labor input, industrial scale, foreign direct investment (FDI), government intervention, financial development, and the level of opening-up on the intelligentization of the manufacturing industry. The specific model is as follows:

$$Y_{it} = \alpha_0 + \alpha_1 L_{it} + \alpha_2 FDI_{it} + \alpha_3 I_{it} + \alpha_4 F_{it} + \alpha_5 G_{it} + \alpha_6 O_{it} + \varepsilon_{it} \tag{7}$$

where $Y$ is the intelligentization level of the manufacturing industry, while $L_{it}$, $FDI_{it}$, $I_{it}$, $F_{it}$, $G_{it}$, $O_{it}$ respectively represent the labor input, FDI, industrial scale, financial development, government intervention, and the level of opening-up. While $\alpha_0$ is a constant, $\alpha_1$, $\alpha_2$, $\alpha_3$, $\alpha_4$, $\alpha_5$, $\alpha_6$ are the regression coefficients. The stochastic perturbation term is given by $\varepsilon_{it}$.

## 3. Indicators Selection and Variable Description

### 3.1. Indicators Selection

According to the relevant concepts of intelligent manufacturing, this paper divides the level evaluation of intelligent manufacturing into three parts: intelligent profit, intelligent innovation, and intelligent equipment. Intelligent profit is the goal of the intelligent manufacturing industry. Whether the intelligent manufacturing industry can play its due role and accelerate the development process of the manufacturing industry mainly depends on products market profitability. Intelligent profit requires the manufacturing industry to be able to timely and quickly obtain customer demands in the process of intelligentization, and to provide personalized products to consumers according to different needs, so as to improve corporate profits. In this paper, the sales revenue of new products is used to measure the level of intelligent profit [20]. Innovation is the fundamental driving force for the continuous development of enterprises. In the process of intelligent manufacturing, intelligent innovations play pivotal roles. Only with good innovation ability can provide guarantee for intelligent equipment and intelligent profit. This paper measures the number of valid invention patents owned by industrial enterprises [2,21]. As the core function of manufacturing enterprises is production, the equipment used in production is paramount and is thus a key step to realize intelligent manufacturing. Only by constantly improving equipment and increasing equipment utilization make an enterprise have more competitive advantages in the market, and only by applying intelligent equipment to production make it realize its value. This paper measured the internal expenditure of R&D [18]. The specific index system is listed in Table 1.

**Table 1.** Evaluation indices of the intelligence level of manufacturing industry in the YREB.

| Primary Indicators | Secondary Indicators | Unit |
|---|---|---|
| intelligent profit | New product sales revenue | $10^4$ RMB |
| Intelligent innovation | The number of patents | % |
| intelligent equipment | Internal expenditure on R&D | $10^4$ RMB |

Note: RMB is renminbi, Chinese currency.

### 3.2. Variable Description

Natural for this research, the manufacturing industry the level of intelligentization is set as the response variable and is evaluated comprehensively from three aspects: intelligent innovations, intelligent equipment, and intelligent profit. Explanatory variables include

labor input, FDI, government intervention, industrial scale, financial development, and level of opening to the outside world. Labor input is an important factor in the development of manufacturing industry, which has an important impact on the research capacity and production efficiency of manufacturing industry. In this paper, the average wage of manufacturing industry is used to measure labor input [22,23]. FDI is an important way for manufacturing industry to introduce technology and capital. It is difficult to effectively promote the development of manufacturing industry only by relying on rich resource. The spillover of knowledge and technology generated by FDI is an important factor for the development of manufacturing industry in the early stage [20]. The government promotes the upgrading of enterprises' equipment by means of fiscal allocation, special support, tax exemption and so on, so as to improve the intelligent level of manufacturing industry. Therefore, the proportion of government's general public budget expenditure in GDP is used to represent government intervention [24]. In the manufacturing industry, large and medium-sized manufacturing enterprises can obtain sufficient financial support in terms of cost input, so as to have a strong research ability, which has a great impact on the intelligence of the manufacturing industry. In this paper, the proportion of large and medium-sized industrial enterprises is used to represent industrial scale [20,25]. Capital is one of the important factors of economic development, especially for the manufacturing industry with a huge capital demand. The more developed the financial industry is, the more it can provide financial support for the R&D investment and technology introduction of the manufacturing industry. Simultaneously, it can reduce operational risks. In this paper, financial development is represented by the total amount of deposits and loans of financial institutions [26]. Since China's reform and opening up, the opening level of the provinces in the YREB has been constantly improved, and the exchange with other countries has been increasingly close. The arrival of advanced knowledge and technology has had an important impact on the intelligentization of the manufacturing industry. In this paper, the total imports and exports were used to assess the level of opening-up with the outside world [27].

Statistics describing the above variables were derived from the Statistical Yearbook of All Provinces, Statistical Yearbook of China Industry [28] and are presented in Table 2. Statistical descriptions are provided in Table 3. The data ranges over the eleven-year period from 2008 to 2018. Due to the existence of missing values in the data, interpolation is used to process and fill these gaps.

**Table 2.** Variable description.

| Variable | Specific Indicators | Unit |
|---|---|---|
| Labor input | Per Capita Wage in Manufacturing | RMB |
| FDI | Foreign direct investment | $10^{12}$ RMB |
| Industrial scale | Government general public budget expenditure as a proportion of GDP | % |
| Financial development | Proportion of large and medium-sized industrial enterprises in industrial enterprises above designated size | % |
| Government intervention | Total deposits and loans of financial institutions | $10^{12}$ RMB |
| Open Level | Total import and export | $10^{12}$ RMB |

**Table 3.** Statistical description in this paper.

|  | Sample Size | Min | Max | Mean | Standard Deviation |
|---|---|---|---|---|---|
| Levels of Intelligentization | 121 | 0.0237 | 8.2865 | 0.8188 | 1.3147 |
| Labor input | 121 | 1.7643 | 32.3131 | 4.5941 | 2.9279 |
| FDI | 121 | 1.74 | 357.60 | 102.8901 | 77.2324 |
| Industrial scale | 121 | 38.79 | 76.83 | 61.9219 | 8.5935 |
| Financial development | 121 | 10.29 | 40.06 | 22.1729 | 7.2257 |
| Government intervention | 121 | 0.8306 | 25.5437 | 7.1629 | 5.4152 |
| Open level | 121 | 0.2307 | 66.4043 | 13.5287 | 18.42019 |

## 4. Results

### 4.1. Comprehensive Analysis of Intelligentization in the YREB

As clearly observable in Figure 1, over the eleven-year period ranging from 2008 to 2018, the level of intelligentization rose in the YREB, though this rise is most notable in intelligentization innovations after 2012. It is suggested that this could be due to the expansion of the scale and quality of Internet services, which accelerates the acquisition and mastery of knowledge, thereby increasing regional expertise. Simultaneously, with the introduction of policies to encourage innovation, for example with reference to the 2015 industrial "Made in China 2025" policy, and the 2015 Opinions of the General Office of the State Council on Promoting the Development of "Internet plus Health Care" document, this steep rise can be explained [5,29].

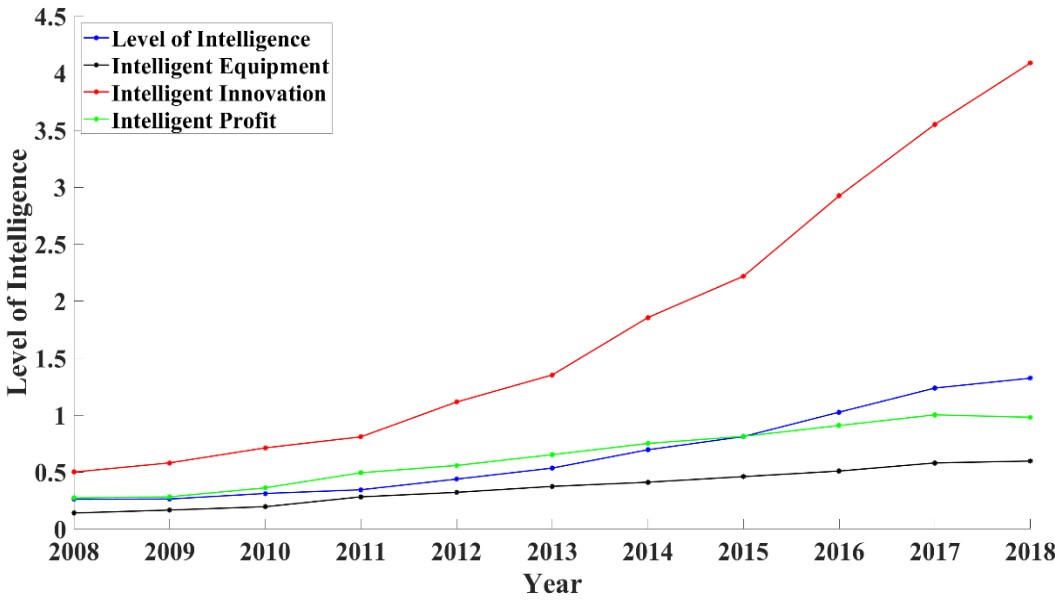

**Figure 1.** Level of intelligentization of the Yangtze River Economic Belt from 2008 to 2018.

However, it can also be observed that both intelligent equipment and intelligent profit have failed to rise as steeply as intelligent innovations. Here, the rising level of innovation failed to play a role in driving core technologies and equipment and thereby exposed a weakness in the YREB manufacturing industry. Moreover, as intelligent equipment is the embodiment of intelligent manufacturing and is increasingly being used to replace humans in a variety of production activities, its presence in the manufacturing industry is essential. However, it is dependent on information, Internet of Things, and communication technology, in addition to other infrastructure [30]. In the eastern YREB, infrastructure to support these activities are present and mature and is a result of large early-stage investments into intelligent equipment and the corresponding support facilities and expertise. This led

and is conducive to stable industry development with a complete industrial chain. In the central and western regions, however, relatively weak infrastructure prevails, though some improvements in infrastructure and intelligent equipment have been made [13].

### 4.2. Levels of Intelligentization

Shown in Figure 2, and as expected, there are discrepancies in the levels of intelligentization in the manufacturing industries of each province or municipality. While each subregion has increasing levels of intelligence in their manufacturing industries, it can however be readily noted that Anhui (after 2010), Jiangsu, Shanghai, and Zhejiang (after 2010) form a cluster away from the other provinces and municipalities. The reason for the high values of intelligentization for Shanghai and Zhejiang could perhaps be traced to the observation that they have been at the forefront of Chinese national economic development since 2008. Additionally, they have strong intelligentization innovation ability and relatively solid foundations of intelligent equipment [31]. Secondly, Anhui and Jiangsu have developed rapidly in terms of intelligentization, which should be attributed to the fact that two provinces are regions where advanced manufacturing industries gather and have abundant resources. Although the rest of the YREB has a low level of intelligentization, it is also growing steadily.

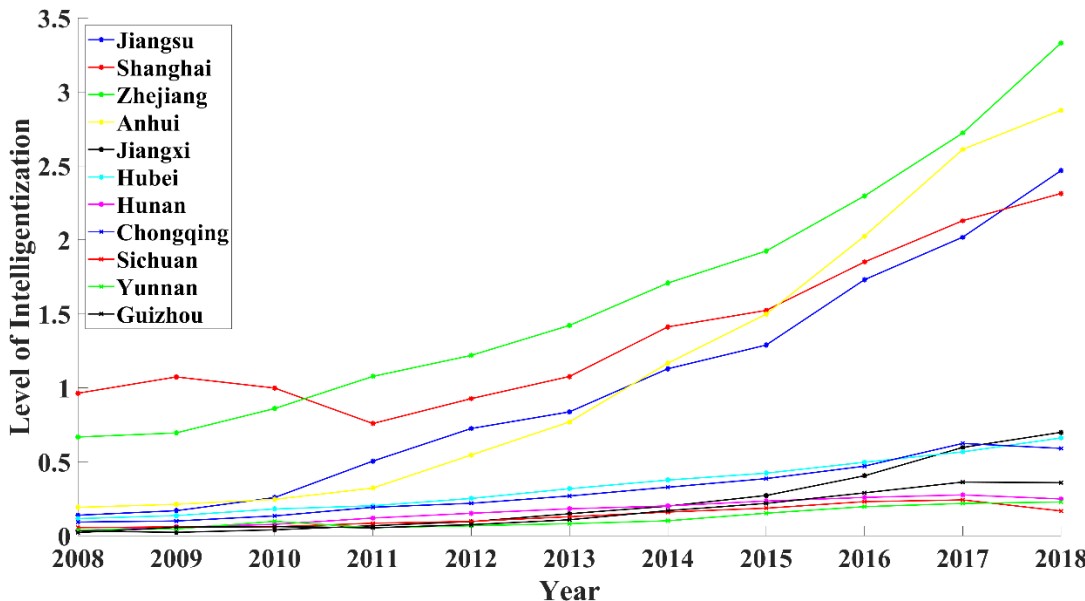

**Figure 2.** Level of intelligentization for each province/municipality over the 2008–2018 eleven-year period.

In Figure 3, the spatial distribution of the YREB manufacturing industry comprehensive intelligence score is given in five-year increments from 2008 (Figure 3a), 2013 (Figure 3b), and 2018 (Figure 3c) to track its temporal evolution. It must, however, be noted that the level of intelligentization is a relative index, and to the best of the author's knowledge, no standard classification currently exists for the level of intelligentization. In Figure 3a, we can see that out of the nine provinces and two municipalities, only Shanghai and Zhejiang achieved a high level of intelligence in its manufacturing industry, relegating the remaining provinces to a low-level classification. It belongs to the coastal areas, where foreign trade is relatively frequent and where advanced technology can be introduced and investment can be obtained.

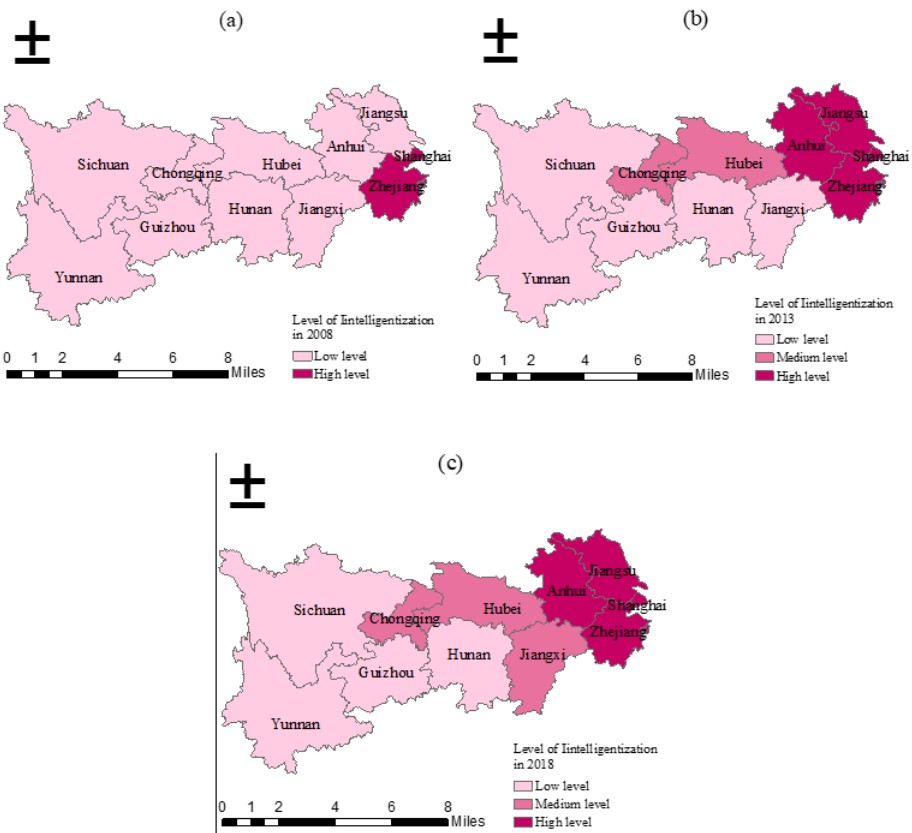

**Figure 3.** Spatial distribution of the manufacturing industry comprehensive intelligence score of the Yangtze River Economic Belt over the 2008–2018 eleven-year period in (**a**) 2008, (**b**) 2013, and (**c**) 2018.

Over a five-year period, however, the number of provinces classified by a high level of intelligentization increased by two (Anhui, Jiangsu), leaving only Shanghai and Zhejiang with their original classifications (Figure 3b). This is perhaps since these two provinces are regions where advanced manufacturing industries are concentrated, and they have abundant resources, which can provide strong support for the intelligent development of manufacturing industries. Furthermore, Chongqing and Hubei have risen from low to medium levels. Guizhou, Hunan, Jiangxi, Sichuan, and Yunnan provinces are still at a low level of intelligentization in their manufacturing industries.

From Figure 3c representing the state of the comprehensive intelligence score of the region in 2018, it can be observed that Anhui, Jiangsu, Shanghai, and Zhejiang were all classified as areas with high levels of manufacturing industry intelligence, and Jiangxi was reclassified as medium level, while Hubei and Chongqing maintained their status as areas with moderate levels of manufacturing industry intelligence. Moreover, it should also be noted that throughout the entire study period, the provinces of Guizhou, Hunan, Sichuan, and Yunnan never achieved a medium- or high-level classification. This is perhaps due to the fact that these areas are relatively poor in natural and human resources, with inadequate infrastructure, and slow economic development [20,32].

While those areas characterized as possessing medium levels of intelligentization in their manufacturing industries are relatively rich in resources and are provided with sufficient financial support, their intelligentization processes have just started and are in the process of accelerating. Additionally, although there has been progress made in intelligent innovation abilities, the transformation of intelligent profits remains sluggish [12,30].

Based on the above analysis, the intelligence level of the manufacturing industry in all provinces of the YREB is not balanced, presenting a trend of "higher in the east, lower in the west", with significant regional gradient differences. Based on improving the intelligence of each province by relying on its own resources, each province should

strengthen inter-provincial exchange and cooperation. Based on comprehensively improving intelligentization, the gap between provinces should be gradually narrowed to promote the balanced development of regional intelligent manufacturing.

### 4.3. Empirical Results Analysis

To conduct an empirical analysis, a brief test of panel data under both fixed and random effect model conditions was done. We found that the fixed effect model was more robust following Hausman test results that returned $p(0.0178)$, which is less than the threshold value of 0.05 and was consequently used in the regression analysis (Table 4).

**Table 4.** Regression results of panel data.

| Variable | Level of Intelligentization | Intelligent Devices | Intelligent Innovation | Intelligent Profit |
|---|---|---|---|---|
| Labor input | 0.0091 | 0.0002 | 0.0145 | 0.0038 |
| | (0.5527) | (0.5970) | (0.3979) | (0.6403) |
| FDI | −7.9040 *** | −0.0898 *** | −15.7026 *** | −0.7750 |
| | (−4.4269) | (−2.2327) | (−4.0044) | (1.1961) |
| Government intervention | −6.7491 *** | −0.0840 * | −14.9401 *** | −1.2899 |
| | (−2.4131) | (−1.6333) | (−2.4322) | (−1.2709) |
| Industrial scale | 0.7511 | 0.0308 * | 3.1947 | 0.6884 ** |
| | (0.6674) | (1.4871) | (1.2925) | (11.1655) |
| Financial development | 0.2507 *** | 0.0051 *** | 0.5583 *** | 0.0853 *** |
| | (9.7382) | (10.8208) | (9.8740) | (9.1320) |
| Open level | 3.3827 ** | 0.1310 *** | 4.3918 | 1.7077 *** |
| | (1.8729) | (3.9425) | (1.1072) | (2.6055) |
| $R^2$ | 0.8866 | 0.9538 | 0.8668 | 0.9399 |

Note: (1) ***, **, and *, respectively, mean that the variable coefficient has passed the significance test at the level of 1%, 5%, and 10%. (2) The value in brackets is the $t$ value.

The above results show that labor input can promote the level of manufacturing intelligentization, but it is not significant. The intelligence level of the manufacturing industry in the YREB is still in the primary stage. Development remains labor- and capital-intensive, with low manufacturing costs being the main driving force. As compared with the primary labor input, the proportion of high-level labor input is relatively small. However, with the intelligent transformation of the manufacturing industry, labor levels have changed, with the labor input changing from primary to senior and professional labor inputs [19,33]. The economically developed provinces and municipalities of Anhui, Jiangsu, Shanghai, and Zhejiang all have high living standards and have attracted a large influx of expertise, especially those who are relatively senior in their positions. Central and western regions have failed to attract and retain this kind of high-quality expertise, leading to a lack of experts in enterprises. Differences in eastern and western region demands in expertise ensure that investments of labor input have little effect on the intelligentization of manufacturing in the YREB. However, it has been noted that in general, increasing the investment in labor input can promote intelligent transformation of the manufacturing industry in the YREB at the present stage [2].

Additionally, it has been suggested that FDI has a significant negative influence on the intelligentization of the manufacturing industry in the YREB. China is regarded as the "world factory" by developed countries, providing them with low-end products or intermediate products. FDI mainly invests in labor-intensive and resource-intensive industries, which have low technological content and low intelligence of corresponding production equipment. Secondly, during the past two decades of manufacturing development in the YREB, the vast majority of manufacturing enterprises rely on technology spillover and lack the ability to independently innovate. When the knowledge and technology spillover brought by FDI is sufficient to support the development of manufacturing industry, fewer enterprises carry out intelligent innovation, and their corresponding intelligent innovation ability is relatively low. In general, the intelligent devices brought by FDI cannot meet

the needs of the intelligent transformation of the manufacturing industry at the present stage and inhibit the ability of intelligent innovation, which leads to the difficulties of the intelligent transformation of the manufacturing industry [5,20].

At the 1% level, government intervention has a significant restraining effect on the level of manufacturing intelligentization. Although there are inherent weaknesses and defects in the market economy, relying solely on the market economy to guide industrial development will lead to market failure. Reasonable intervention of the government can make up for the weaknesses and defects of the market economy and ensure the healthy development of the market economy. However, it should be noted that the manufacturing industry in the YREB has sufficient capital strength and professional expertise to support the intelligent transformation of the manufacturing industry. Government fiscal spending can promote manufacturing equipment and improve productivity, but these activities may disrupt the market and lead to an excessive reliance on the government and decrease the desire to innovate. None of these are conducive to the independent development of the manufacturing industry [34,35]. Although not significant, the industrial scale has a positive influence on the intelligentization of the manufacturing industry. The first reason is that the industrial scale can promote the exchange and cooperation of knowledge among industries and enable each industry to carry out the activities with the most production advantages through specialized division of labor, so as to improve industrial productivity and reduce production costs [5]. Secondly, the industrial scale can promote improvement in innovation. The spillover effect of knowledge in certain industries can promote innovation in related industries so as to enhance overall innovation. Moreover, the industrial scale helps enterprises to complement each other within the industry, so as to establish a good brand image, so that high-quality products can be spread outside the regional market [20,31]. In general, the industrial scale is conducive to improving industrial innovation ability, reducing production costs, and expanding the product market, which are the basic elements of industrial intelligent transformation.

Financial development significantly promotes the improvement of intelligentization at the 1% level. The development of the financial industry can accelerate the speed of regional capital accumulation and promote the rapid development of various industries in the region. The intelligent transformation of the manufacturing industry requires a large amount of funds to be invested in enterprise R&D, equipment upgrading, and other activities, which are risky. In order to maintain the normal production and operation activities of the enterprise, the enterprise will perhaps not invest a large amount of working capital in the process of intelligent transformation. The development of the financial industry can effectively disperse the risks of enterprises to ensure that enterprises have enough working capital to meet their daily needs and can obtain financial support to purchase intelligent equipment, so as to promote the intelligent transformation of enterprises [36,37]. The level of opening-up has a significant positive influence on the level of intelligentization in the manufacturing industry. The improvement of the level of opening-up can make each enterprise access the latest information, in addition to leading to a variety of experience and knowledge collision and integration, so that each enterprise constantly improves innovation capabilities. Through export, the manufacturing industry can expand production scales and generate economies of scale. Through the import of advanced equipment and technology, manufacturing industry production processes and R&D of core industrial technology can be promoted. Improvements to the level of opening-up can generate more economic profits to small- and medium-sized enterprises to allow for greater development funds, and thus this can be translated into improving innovation, increasing the overall level of intelligentization in the YREB manufacturing industry [27,38,39].

## 5. Conclusions

This paper evaluated the level of intelligentization about the manufacturing industry in the YREB from the aspects of intelligent innovations, intelligent equipment, and intelligent profits. By measuring the intelligent level of provinces in the YREB from 2008 to

2018, the rationality of the intelligent index system is verified, which provides reference for the construction of the intelligent evaluation system of the regional manufacturing industry. Secondly, empirical research shows that FDI, financial development, government intervention, and the level of opening-up have significant impacts on the level of intelligentization about the manufacturing industry in the YREB, while labor input and industrial scale have no significant impact on the level of intelligentization about the manufacturing industry, which further confirms the research conclusion of Li et al. [5,20].The conclusions are as follows:

(1) The level of intelligentization about the manufacturing industry in the YREB is generally on the rise, among which intelligent innovation is significantly higher than intelligent equipment and intelligent profits. The research shows that in the process of manufacturing intelligence in the YREB, the speed of intelligent innovation continues to accelerate, and the core technology continues to improve, which lays a solid foundation for the improvement of the level of intelligent equipment. The moderate advance of intelligent innovation is not only conducive to the improvement of the intelligent profits of manufacturing industry, but also conducive to guiding the development of intelligent equipment.

(2) The level of intelligentization about the manufacturing industry in the YREB is not balanced, with significant gradient difference. By 2018, Shanghai, Jiangsu, Zhejiang, and Anhui were in the first echelon of intelligentization about the manufacturing industry in the YREB, Chongqing, Hunan, and Jiangxi were in the second echelon, and the rest were in the third echelon.

(3) Empirical research shows that financial development and the level of opening-up significantly promote the level of intelligentization about the manufacturing industry in the YREB, government intervention and FDI can significantly inhibit manufacturing intelligence, while labor input and industrial scale have no significant impact on manufacturing intelligence.

Finally, in view of the problems existing in the process of intelligentization about the manufacturing industry in the YREB, this paper suggests that improving the independent innovation ability is the key to the intelligent development of the YREB and paying attention to the integrated development of other industries and gradually realizing intelligentization according to regional characteristics are the good way to the intelligent development of the YREB. The following suggestions are put forward:

(1) Closer attention should be paid towards the cultivation of innovative experts and mastering the art of independent innovation. The intelligent transformation of the manufacturing industry in the YREB needs to be deeply integrated with modern information technology and network technology. Manufacturing experts should have multidisciplinary knowledge and strong innovation abilities. Because the current mode of expertise training can no longer meet the needs of intelligent development of the manufacturing industry, training should be diversified and specialized. Enterprises should strengthen cooperation with universities and research institutes, for the joint training of prospective employees, and provide intellectual support for the intelligent transformation of manufacturing industry by promoting the research of key topics and the implementation of key projects.

(2) Improve the financial system and reduce the risk of enterprise transformation. The development of the financial industry can provide financial support for the intelligent transformation of the manufacturing industry. In terms of policies, the financing cost of manufacturing enterprises can be reduced by optimizing the credit structure. Financial policies should be strengthened to support intelligent enterprises and reduce their transformation risks, so as to promote the intelligent transformation of the manufacturing industry.

(3) Integrated development amongst industries should be encouraged and based on regional characteristics, and the intelligentization of the manufacturing industry promoted. Traditionally, there is a systemic lack of horizontal communication among

various YREB industries with a low proportion of innovation achievements of productive forces. The lack of synergetic innovation capacity restricts the intelligent transformation of the manufacturing industry in the YREB. Compared with the traditional manufacturing mode, manufacturing intelligentization relies on the development of the Internet and producer services among other facets of the process. The mutual integration of industries it is not only conducive to the improvement of the level of the manufacturing intelligentization but also promotes the development of other industries. Because the YREB stretches across the eastern and western parts of China, due to different levels of intelligentization in each region, appropriate development strategies should be formulated according to the actual conditions of each region.

This work is limited in that it only considered three aspects of the manufacturing industry's level of intelligentization: intelligent innovation, equipment, and profit. Noting that there are many other factors, the index selection system of this research could thus be improved if other facets of intelligentization in the manufacturing industry were considered. Additionally, due to the availability and completeness of data, the selection of some variables such as human capital input may be biased. The selection of measurement indicators relied heavily on objective rather than subjective indicators. It is currently unknown what effects the inclusion of subjective indicators would have on the results here, and thus future research should investigate the robustness of this paper through their inclusion. While the manufacturing industry covers over 30 subsectors that include the textile, pharmaceutical, and textile industries, this paper only considered the intelligentization of the entirety of the manufacturing industry and ignored each subsector's characteristics. Macroscopic influencing factors on manufacturing intelligentization in the YREB from FDI, government intervention, and financial development were studied in this paper, but microscopic influencing factors on a given enterprise were ignored. Microscopic influences should be included in the research, and intelligent influencing factors should be explored according to the characteristics of different industries.

**Author Contributions:** D.T., L.W. and B.J.B. contributed equally to this work. All authors have read and agreed to the published version of the manuscript.

**Funding:** This research was funded by The Open Fund of the China Institute of Manufacturing Development, Nanjing University of Information Science and Technology, P.R. China, grant number (SK2020-0090-11).

**Institutional Review Board Statement:** Not applicable.

**Informed Consent Statement:** Not applicable.

**Data Availability Statement:** The datasets generated during and/or analyzed during the current study are available in the State Statistical Bureau, China Statistical Yearbook 2008–2018 repository (https://data.stats.gov.cn/, accessed on 12 June 2020).

**Acknowledgments:** We thank the editor and reviewers for careful review and insightful comments.

**Conflicts of Interest:** The authors declare no conflict of interest.

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
