# Peer review of "An Evaluation of the Yangtze River Economic Belt Manufacturing Industry Level of Intelligentization and Influencing Factors: Evidence from China"

_sustainability, doi:10.3390/su13168913_

Round 1

Reviewer 1 Report

Manufacturing Industry Level of Intelligentization and Influencing Factors: From China.

Comments and suggestions are the following:

  1. What does mean "intelligent manufacturing sector". The one definition in the introduction: “Briefly, intelligent manufacturing involves the digitization, informatization and networking of manufacturing industry, which is a general concept covering a series of topics (Liu et al., 2019)” is not sufficient. Please express the areas of application for intelligent manufacturing. Please revise the structure of your theoretical background section. You should present the findings from different authors and please do not miss to implement the research model as well as a research gap.
  2. Why has data been collected since 2008?
  3. What is the difference between the “intelligentization of the manufacturing industry” and "intelligent manufacturing"?
  4. Why the entropy weight method was chosen?
  5. On what kind of basis were such indicators selected: the influence of labor input, industrial scale, foreign direct investment (FDI), government intervention, financial development, and the level opening-up on the intelligentization of the manufacturing industry?
  6. Model 7 - why is it a linear relationship?
  7. The research sample must be described in detail
  8. How the findings can be exploited by future similar works?
  9. Please add the discussion - e.g. please define more precisely and detailed the additional contribution of the research to the recent state of the research field an also add the limitations of the work.

Reviewer 2 Report

I have carefully reviewed the manuscript, titled as “An Evaluation of Yangtze River Economic Belt Manufacturing Industry Level of Intelligentization and Influencing Factors: From China”. This study mainly investigates the level of intelligentization for manufacturing industry and influential factors. I think this paper presents an interesting empirical analysis by using provincial panel data.

The main limitation of the research is whether the measures used for variables are appropriate or not. For example, the average wage of manufacturing industry is measured for labor input in this paper. However, it is unclear if this better reflects the labor input for the model because the model intends to compare intelligentization of manufacturing of different provinces. Different measures, such as total labor cost or wage expenditure, might be proper.

The authors are advised to check other variables, such as foreign direct investment (FDI) and government intervention. The fact that FDI has a significant negative impact sounds very counter-intuitive. If FDI turned out to be insignificant, what the authors discussed in Section 4.3 would make sense. However, if the factor’s effect is significantly negative (even with alpha=0.01), the explanation from Line 336 to 350 may not be enough. The authors are advised to check whether the selection of variable or data is appropriate.

The authors could do the same thing for government intervention to check whether “proportion” of government’s “general public” expenditure would make sense or not as a measure for the factor. For instance, the authors can consider a government direct subsidy for manufacturing firms instead.

In addition, there are lots of language problems such as errors in the use of the articles a/an/the, the use of plural/singular, wrong sentences, etc. The authors are advised to have their manuscript professionally reviewed and edited.

For example, grammatical errors include:

  • Line 11: over the recent decades à Over the recent decades
  • Line 11: fields manufacturing, à fields of manufacturing (without comma)
  • Line 17: applied to three evaluation criteria: intelligent innovation, intelligent equipment and intelligent profit.
  • Line 19: the level intelligence à the intelligentization level or the level of intelligentization
  • Line 61: Selected panel data à selected panel data
  • Modify the entire sentence in Line 60
  • Modify the entire sentence in Line 72
  • Modify the entire sentence in Line 80
  • Modify the entire sentence in Line 97

Finally, the authors are advised to check the following additional comments below:

  1. Terms are mixed. For example, both Intelligent innovation and intelligence innovation are used at the same time in the manuscript.
  2. Based on Eq. (1) and (2) in Step 1, which X_ij is used in Eq .(3)? Is it positive or negative indicator?
  3. In Line 151, the authors say “This paper mainly explores the influence of labor input, industrial scale, foreign direct investment (FDI), government intervention, financial development, and the level opening-up on the intelligentization of the manufacturing industry (Table 1).” However, Table 1 is summary of indices rather than factors.
  4. In Table 1, does 104 mean 10^4?
  5. No proper explanation how FDI is measured. What is the unit of the variable? The authors need to include units of variables in Table 2.
  6. Data sources for the statistics in Line 218 to 220 must be included in References so that readers can verify its validity.
  7. Not sure how the authors decide “high” and “low” levels for Figure 3a – 3c. This should be explained in relation to Figure 2. Based on the explanation, it seems that the level of intelligentization shown in Figure 2 and 3 is relative rather than absolute. The authors can mention it in Section 2.1 for instance.
  8. Figures must be substituted by the one with highest resolution. For instance, it’s hard to recognize letters in Figure 3.

If the authors can address these issues I would be willing to look at a revised version.

Round 2

Reviewer 1 Report

The authors have significantly improved the text given in the manuscript, however I have still comments:

  1. Please define more precisely and detailed the additional contribution of the research to the recent state of the research field.
  2. Please add the limitations of the work.

Reviewer 2 Report

Check Eq. (1): there are two parentheses in the equation.

In Figure 2, it should read "Jiangsu" rather than "Levels of Jiangsu".
